# UFL1 Alleviates LPS-Induced Apoptosis by Regulating the NF-κB Signaling Pathway in Bovine Ovarian Granulosa Cells

**DOI:** 10.3390/biom10020260

**Published:** 2020-02-09

**Authors:** Xinling Wang, Chengmin Li, Yiru Wang, Lian Li, Zhaoyu Han, Genlin Wang

**Affiliations:** College of Animal Science and Technology, Nanjing Agricultural University, Nanjing 210095, China; 2017105030@njau.edu.cn (X.W.); T2014046@njau.edu.cn (C.L.); 2017205015@njau.edu.cn (Y.W.); zyhan6708@njau.edu.cn (Z.H.); glwang@njau.edu.cn (G.W.)

**Keywords:** apoptosis, bovine granulosa cells (bGCs), UFL1, lipopolysaccharide (LPS)

## Abstract

Ubiquitin-like modifier 1 ligating enzyme 1 (UFL1) is an E3 ligase of ubiquitin fold modifier 1 (UFM1), which can act together with its target protein to inhibit the apoptosis of cells. Lipopolysaccharides (LPS) can affect the ovarian health of female animals by affecting the apoptosis of ovarian granulosa cells. The physiological function of UFL1 on the apoptosis of bovine (ovarian) granulosa cells (bGCs) remains unclear; therefore, we focused on the modulating effect of UFL1 on the regulation of LPS-induced apoptosis in ovarian granulosa cells. Our study found that UFL1 was expressed in both the nucleus and cytoplasm of bGCs. The results here demonstrated that LPS caused a significant increase in the apoptosis level of bGCs in cows, and also dramatically increased the expression of UFL1. Furthermore, we found that UFL1 depletion caused a significant increase in apoptosis (increased the expression of BAX/BCL-2 and the activity of caspase-3). Conversely, the overexpression of UFL1 relieved the LPS-induced apoptosis. In order to assess whether the inhibition of bGCs apoptosis involved in the nuclear factor-κB (NF-κB) signaling pathway resulted from UFL1, we detected the expression of NF-κB p-p65. LPS treatment resulted in a significant upregulation in the protein concentration of NF-κB p-p65, and knockdown of UFL1 further increased the phosphorylation of NF-κB p65, while UFL1 overexpression significantly inhibited the expression of NF-κB p-p65. Collectively, UFL1 could suppress LPS-induced apoptosis in cow ovarian granulosa cells, likely via the NF-κB pathway. These results identify a novel role of UFL1 in the modulation of bGC apoptosis, which may be a potential signaling target to improve the reproductive health of dairy cows.

## 1. Introduction

The reproductive performance of dairy cows has attracted growing attention in the cattle industry [1,2]. The ovaries are a key reproductive organ and play a crucial role in the animal production performance. Follicles are a basic constituent in the ovary, which is mainly composed of a single type of oocyte, named granulosa cells (GCs). With the proliferation of the granulosa cells, the follicle goes through several major phases and eventually matures to the point of ovulation [3,4]. Previous studies have indicated that follicular granulosa cells modulate follicular development through excreting estrogen and progesterone [5]. Granulosa cells, as key functional components of follicles, perform an initiating role in follicular atresia and further affect the reproductive facility [6,7,8]. Granulosa cell apoptosis appears to be an essential part of ovarian development, and it is a reflection of the mitogenic growth of the follicle [9]. Previous studies have elaborated that lipopolysaccharides (LPS) could induce an ovarian inflammatory response and affect ovarian development directly [10,11].

In recent years, the diseases caused by upper genital tract infections of dairy cows are on the rise; they have greatly damaged the life-long benefits of dairy cattle in reproduction [12,13]. Gram-negative bacteria are one of the most important bacteria in cow upper genital tract infections. The LPS released by these bacteria can cause great damage to the body at a certain concentration [14]. In addition, Gram-negative bacteria are also common in the rumen and intestines. Subacute acidosis or damage of the intestinal epithelial mucosa system will also cause an increase in LPS content [15]. LPS can diffuse into the blood and even into the follicular fluid through damaged epithelial mucosa of the uterus and rumen, thus affecting the normal development of the ovary [16]. LPS has been proven to trigger inflammation and follicular apoptosis in rat ovarian granulosa cells [17]. Previous studies have shown that LPS causes inflammation and cell apoptosis via downregulating the expression of BCL-2 and elevating the levels of BAX and caspase-3 (CAS3) [18,19]. Cell apoptosis has a close association with inflammation in a host. Therefore, maintaining the normal apoptotic level of follicular granulosa cells is essential for the reproductive performance of dairy cows.

UFL1 (ubiquitin-like modifier 1 ligating enzyme 1), was recently proven to be an important regulator of pathological and physiological cellular processes [20,21], which can be detected in the cytoplasm, nucleus, and endoplasmic reticulum [20]. Studies have proven that UFL1 plays a crucial role in cell homeostasis and inflammation response [22,23]. UFL1 also plays an important role in the regulation of nuclear factor-κB (NF-κB) signaling [24]. As shown earlier and confirmed in our previous studies, UFL1 alleviates mastitis by inhibiting the NF-κB signal pathway [25]. An inflammatory response causes an increased level of apoptosis. Moreover, Lemaire et al. found that knocking out UFL1 causes endoplasmic reticulum stress and induces an unfolded protein response, which is closely related to apoptosis [14]. Studies have shown that the activation of the NF-κB signaling pathway, induced by LPS, promoted apoptosis [26,27]. Given that UFL1 can regulate inflammation and inhibit the activity of the NF-κB transcription factor, these findings raise the question of whether UFL1 has a homeostatic role to regulate the apoptosis level by inhibiting LPS-induced activation of the NF-κB pathway. Based on this assumption, in the present study, we evaluated the role of UFL1 on apoptosis in granulosa cells in an LPS-induced injury cell model with NF-κB. This study has great significance for animal follicular development and reproduction and dairy industry development.

## 2. Materials and Methods

### 2.1. Cell Culture

The cow ovaries were taken from a nearby slaughterhouse, stored in physiological saline at 37 °C and shipped to the laboratory within 2 h. Excess tissues were removed with sterile forceps and scissors, and then the ovaries were washed several times with pre-warmed saline. We used a sterile syringe to extract follicular fluid into 15 mL centrifuge tubes; the follicular fluid was extracted from the small (3–6 mm in diameter) follicles. Following centrifugation at 1000× *g* for 5 min, the cells were re-dissolved in 10% fetal bovine serum in culture medium (all were from Gibco Life Technologies, Carlsbad, CA, USA) under a 37 °C humidified incubator. The process of the test met the requirements of the Experimental Animal Welfare and Ethics Committee of Nanjing Agricultural University.

### 2.2. Cells Treatment

Our study was composed of three parts. The first part involved the control group and control + LPS group; on this basis, the second part was set up as three parallel groups, namely NC (negative control), NC + LPS, and siUFL1 (small-interfering *Ufl1*, which knocks down the *Ufl1* gene) + LPS; similarly, the third part was the control, control + LPS, and UFL1 (*Ufl1* plasmid; the overexpression of *Ufl1*) + LPS.

### 2.3. Cell Viability Assay

The experiment was completed with the Cell Counting Kit-8 (CCK-8; Nanjing Jiancheng, China). The cells were cultured in a 96-well plate at 100 μL per well, at 37 °C and 5% CO_2_, for 24 h to achieve a density of 40%–50% and treated with different concentrations of LPS for 6 h. Then CCK-8 solution was added to the cells (attempting to not to create bubbles in the well) for 1.5–2 h. Finally, we use a microplate reader to detect the absorbance at 450 nm.

### 2.4. Immunofluorescence 

The cells were seeded in a 24-well plate, in which the slides were placed and washed with pre-warmed PBS three times, for 5 min each time, when the cells density was 80%–90%. We fixed the cells with 4% paraformaldehyde solution for 1 h, and then the cells were permeabilized with 0.5% TritonX-100 for 20 min. After that, the cells were washed with PBS and incubated with 2% BSA (bovine serum albumin) for 1 h and incubated with an antibody specific for UFL1 (1:200 Proteintech, Chicago, USA) in a 4 °C refrigerator overnight. After washing with PBS, secondary antibody (1:50) was added to the cells at room temperature, for 1 h (in a dark box to protect from light). After similarly washing with PBS, the nucleus was dyed with DAPI for 10 min. Finally, the cells were washed with PBS three times and observed under a confocal microscope (Zeiss LSM 700 META).

### 2.5. Cell Transfection

We used Lipofectamine 2000 (Invitrogen, Carlsbad, CA, USA) to transfect cells according to the instructions. The plasmid was extracted with glycerol broth (GenePharma, Shanghai, China) according to the instructions, and it was identified by the Jinsirui Bio Company (Nanjing, China). The amplified product was purified and the *Ufl1* gene was cloned into the pEX-3 vector, and it was then transferred into competent cells. Small-interfering RNAs (siRNA) of *Ufl1* were synthesized by Shanghai GenePharma. The bGCs were seeded into six-well plates and transfected at a density of 60%–70%. For overexpression of *Ufl1*, cells were transfected with 30 nM plasmid, and for siRNA knockdown, cells were transfected with 60 nM siRNA (the sequence and antisense sequence of siRNA are 5′-GCAGCAGAAGCUUGUGAUATT-3′, 5′-UAUCACAAGCUUCUGCUGCTT-3′).

### 2.6. Total RNA Extraction and Real-Time Fluorescence Quantitative PCR

Total RNA was extracted with the TRIzol reagent (Invitrogen) (all steps were performed on ice to prevent RNA degradation), RNA concentration was determined with the Nanodrop ultra-micro spectrophotometer, and RNA purity was determined by OD 260/280 and OD 260/230. The OD 260/280 value was between 1.8 and 2.1, indicating good RNA purity. The RNA was reverse transcribed into complementary DNA using Prime ScriptTM RT Master Mix (TaKaRa, Otsu, Japan). Quantitative real-time PCR (qRT-PCR) was performed by SYBR^®^ PremixExTaqTM (TaKaRa, Otsu, Japan), and standard protocols were used on the Applied Biosystems 7500 HT Sequence detection system. The primers were synthesized by Kingsray Biotech (NJ, China). The experimental data were normalized by that of GAPDH and analyzed by the 2-ΔΔCt method.

### 2.7. Western Blot

The protein extraction process was processed on ice. The cells samples in the six-well plate were gently washed three times with PBS, and 200 μL of RIPA lysis buffer (Beyotime, Shanghai, China), containing the protease inhibitor Phenylmethanesulfonyl fluoride (PMSF), was added to each well for 30 min. The cells were transferred to centrifuge tubes, then centrifuged at 15,000× *g* for 10 min, and the supernatant was retained. The protein concentration was measured by a BCA kit (Beyotime, China). We added 20 μL of 5 × SDS loading buffer into 80 μL of the adjusted protein sample, and the mixture was heated in a 100 °C water bath for 10 min. Gel electrophoresis was performed using a 12% strength pre-formed gel purchased from Kingsley, to separate the proteins, and was then transferred to a PVDF membrane. The membrane was blocked with the blocking solution (5% skim milk powder; 0.5% Tris-buffered saline-Tween) prepared in advance for 1 h and was washed three times with TBST for 10 min each time, and the corresponding strips were cut and then placed in a 4 °C refrigerator overnight. After the strips were washed, the secondary antibody was added for 1.5 h, and the PVDF membrane was sufficiently contacted with an enhanced chemiluminescence reagent (Biosharp) and exposed to an enhanced chemiluminescence detection system (Amersham, Piscataway, NJ, China). Optical density analysis was performed using the Image J (National Institutes of Health, Bethesda, MD, USA) software processing system.

### 2.8. Flow Cytometry for Apoptosis

After the cells were transfected for 48 h, the experiments were performed using the Annecin V-Alexa Fluor 647/PI (FcMACS, NJ, China) Apoptosis Assay kit, and the specific procedures were carried out according to the instructions. The cells were washed twice with pre-chilled PBS, and then resuspended with 250 μL of binding buffer (diluted 1:4 with deionized water), 100 μL of cell suspension were added to flow tubes, and 5 μL of Annecin V-Alexa Fluor 647 and 10 μL propidium iodide solution were mixed and incubated at room temperature for 15 min in the dark, and then 400 μL of PBS was added into the reaction tubes. Finally, flow cytometry (FCM) was used to detect the level of cells apoptosis. Data analysis was performed with the Flowjo software version 10.0.7 (Becton, Dickinson and Company, Franklin Lakes, JD, USA).

### 2.9. HE Staining

Ovarian tissue samples were fixed in 4% paraformaldehyde and dehydrated with an ethanol concentration gradient. Tissues were sectioned after paraffin embedding and stained with hematoxylin–eosin (HE) (Sigma-Aldrich, St. Louis, MO, USA). The tissues were observed with an optical microscope (Olympus Corporation, Tokyo, Japan).

### 2.10. Immunohistochemistry

The tissues fixed with polyformaldehyde were embedded in paraffin, washed with PBS three times, and then treated with hydrogen peroxide for 30 min. After that, the tissues were incubated in blocking solution for 1 h, and then incubated overnight with UFL1 polyclonal antibodies (1:400, Proteintech, Chicago, CA, USA). Finally, the samples were incubated in Dolichos biflorus agglutinin (DBA) staining solution after treatment with the second antibody for 1 h. The control group was stained without the first antibody.

### 2.11. Statistical Analysis

The SPSS 20.0 (SPSS Inc., Chicago, IL, USA) was used for statistical analysis and the GraphPad Prism 6.01 software (GraphPad Software Inc., San Diego, CA, USA) was used to generate a graph. It is considered that *p* < 0.05 was a significant difference.

## 3. Results

### 3.1. The Expression of UFL1 in Cow Ovarian Tissue and bGCs

Follicle-stimulating hormone receptor (FSHR) immunocytochemical staining is the specific staining of ovarian granulosa cells, where the nucleus is dark blue, and the FSHR positive staining is localized in the cell membrane, which is stained green. The positive rate of FSHR is more than 95% under the microscope (Figure 1a), which indicates that the purity of the isolated granulosa cells is over 95%, which can be used in subsequent experiments.

To explore the location of UFL1 in ovarian tissue and bGCs, we used immunohistochemistry and immunofluorescence techniques. First, the bovine ovary was analyzed by immunohistochemistry, and UFL1 was expressed in bovine ovarian cells (Figure 1b). In addition, the immunofluorescence results show that UFL1 is expressed in both the nucleus and cytoplasm of bGCs, however, mainly in the cytoplasm (Figure 1c).

### 3.2. The Apoptosis Levels of bGCs under LPS Treatment

CCK-8 was used to determine the effects of LPS on the cytotoxicity to GCs (Figure 2a). The cells were treated with LPS (1 μg/mL) for 6 h, and the expression of apoptosis-related proteins detected at the protein level was found in comparison with the control group. The levels of apoptosis were upregulated in the LPS-treated group (Figure 2b).

### 3.3. The Expression of UFL1 in LPS-Induced bGCs

Using Western blotting and real-time PCR, we detected that LPS treatment significantly increased the mRNA and protein levels of UFL1 in bGCs (Figure 3a,b). We therefore speculate that UFL1 may play a role in the LPS-induced apoptosis of ovarian granulosa cells. To further investigate the role of UFL1, we used small-interfering RNA to knock down the expression of *Ufl1* in bGCs. The results showed that under the conditions of LPS induction, *Ufl1* siRNA can significantly reduce the expression of UFL1, both in mRNA and proteins (Figure 3a,b). Conversely, overexpression of *Ufl1* significantly increases the expression level of UFL1. The results showed that *Ufl1* siRNA and plasmid were effectively regulated in the expression of UFL1 by transfected cells.

### 3.4. UFL1 Regulated LPS-Induced Apoptosis in bGCs

The apoptosis of bGCs was analyzed by flow cytometry. As shown, LPS treatment increased the level of bGCs apoptosis, and the knockdown of *Ufl1* resulted in a significant increase in apoptosis in the LPS-treated granulosa cells (Figure 4a). In contrast, the overexpression of *Ufl1* reduced LPS-induced bGCs compared to the Con + LPS group (Figure 4b).

### 3.5. UFL1 Reduced the Expression Levels of Genes and Proteins Associated with Apoptosis

Consistent with the results of flow cytometry, Western blot (Figure 5c) and real-time PCR (Figure 5a,b) showed that knockdown of *Ufl1* increased the BAX/BCL-2 ratio in LPS-induced cells under LPS treatment conditions. Additionally, the expression level of CAS3 was upregulated. These indicators were decreased compared with the LPS-treated group after overexpression of *Ufl1*. Taken together, these data show that UFL1 can inhibit apoptosis in LPS-stimulated bGCs.

### 3.6. UFL1 Inhibited LPS-Induced Activation of the NF-κB Pathway

Inhibition of the NF-κB signaling pathway may have an inhibitory effect on apoptosis of bGCs. To further confirm this idea, we examined the expression of p65 and p-p65 by Western blot. As shown in Figure 6, LPS activated NF-κB, resulting in a significant increase in p-p65, knockdown of *Ufl1* further increased the phosphorylation of p65, while overexpression significantly inhibited the activation of NF-κB p65 and inhibited the expression of p-p65.

## 4. Discussion

In this study, we evaluated the effects of UFL1 on the apoptosis of bGCs through an LPS-induced cell model. We found that LPS increased the apoptosis of bGCs. UFL1 silencing increased the apoptotic level in granulosa cells under LPS-induced conditions, whereas overexpression of UFL1 protected against apoptosis in LPS-stimulated bGCs. Meanwhile, UFL1 decreased the protein level of NF-κB p-p65, which indicated that UFL1 may regulate apoptosis via the NF-κB pathway. These findings identify a novel role of UFL1 in the modulation of the apoptotic level of granulosa cells.

LPS is a pivotal constituent of the cell wall of Gram-negative bacteria [28]. It has been well established that LPS has a detrimental effect on follicular and luteal growth. Low concentrations of LPS can be found in the body and follicular fluid of normal cows. Previous studies have found that the LPS content of follicular fluid in healthy cattle with normal periodic ovarian activity is very low, with an average of 0.06 ± 0.04 ng/mL [29]. However, the concentration of LPS in the follicular fluid of postpartum cows with female reproductive diseases is about 1000 times higher than that of normal postpartum cows [15]. LPS has been compared to a “double-edged sword”. On the one hand, it can induce cell apoptosis, and on the other hand, it can promote cell proliferation; which role it plays depends on the different experimental subjects and treatment concentrations [30,31]. The establishment of an apoptotic injury model using LPS stimulation is a classic model; however, the difference in treatment concentration is due to the differences in experimental animals, etc. Our results show that for ovarian granulosa cells, 1 μg/mL is the optimal treatment concentration for LPS. On this basis, the treatment of granulosa cells with LPS significantly increased the level of apoptosis, which is consistent with previous studies [32]. To summarize, we confirm that treatment of granulosa cells with a certain concentration of LPS increases the apoptosis of granulosa cells.

Tilly et al. first demonstrated that granulosa cell apoptosis occurred in chickens during the process of follicular atresia and suggested that the essence of follicular atresia is probably due to the apoptosis of granulosa cells [33]. Subsequent studies have shown that the proliferation and apoptosis of granulosa cells are closely related to the follicular developmental potential [34]. UFL1 was proven to be a crucial regulator of cellular stress, which is mainly expressed in the endoplasmic reticulum [35]. In our previous study, we found that UFL1 could possess anti-inflammatory and cyto-protection effects in LPS-induced dairy mammary epithelial cells.

Studies of UFL1′s role in the mammary glands of cows are very common; however, the physiological function of UFL1 protein in cow ovaries has been rarely reported. Our results showed that UFL1 was highly expressed in ovarian granulosa cells and tissues, and the expression of UFL1 was upregulated in granulosa cells after stimulation with LPS. Therefore, we speculated that UFL1 may be related to the apoptosis of granulosa cells induced by LPS. UFL1 silencing significantly increased the level of apoptosis induced by LPS in our results. Conversely, overexpression of UFL1 protected against apoptosis in LPS-stimulated bGCs. These findings indicate that UFL1 is capable of attenuating aggravated apoptotic levels induced by LPS in the bGCs.

NF-κB is widely distributed in the transcription process of cells and participates in a series of life activities, including immune responses and cell apoptosis [36,37,38]. P65 is a key active subunit in NF-κB transcription [39,40]. NF-κB is activated under certain stimulating conditions, and then p65 is phosphorylated to p-p65 in the nucleus [41,42]. Previous studies have shown that UFL1 inhibits the activity of NF-κB and NF-κB-mediated transcription [43]. Therefore, we speculated that UFL1 may control granulosa cell apoptosis induced by LPS, by inhibiting NF-κB activation. In order to confirm this hypothesis, we examined the expression of NF-κB p-p65 by Western blot. Our results indicated that LPS treatment led a significant increase in the NF-κB p-p65 protein, and the knockdown of UFL1 further increased phosphorylation of NF-κB p65 to p-p65, whereas overexpression of UFL1 significantly inhibited the expression of NF-κB p-p65. These results confirmed that UFL1 is indeed involved in the phosphorylation of NF-κB, and that the knockdown of UFL1 promoted phosphorylation of NF-κB p65. Our results agree with those of Li et al. [26]. However, whether UFL1 directly regulates the phosphorylation of NF-κB p65 and the specific mechanism of this effect require further studies.

## 5. Conclusions

UFL1 is widely expressed in cow ovaries, and it was found to alleviate the apoptosis of granulosa cells triggered by LPS. Furthermore, UFL1 inhibits the NF-κB signal pathway, which is manifested by downregulating the level of NF-κB p-p65. This partly explains the regulatory effect of UFL1 on the apoptosis of granulosa cells. These results identify a novel role of UFL1 in the modulation of bovine ovarian granulosa cell apoptosis, which may be a potential signal target to improve the reproductive performance of dairy cows.

## Figures and Tables

**Figure 1 biomolecules-10-00260-f001:**
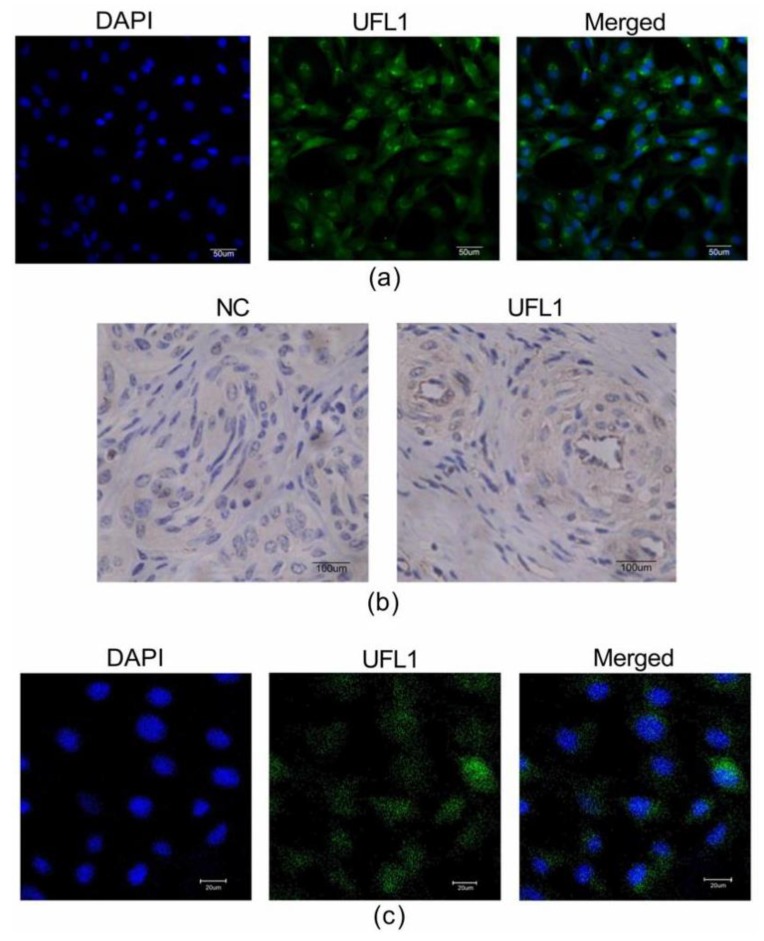
Ubiquitin-like modifier 1 ligating enzyme 1 (UFL1) is expressed in ovarian tissue and the cytoplasm and nucleus of granulosa cells. (**a**) More than 95% of the cells in the isolated cultured cells are ovarian granulosa cells. Identification of ovarian granulosa cells by immunofluorescence. (**b**) Immunofluorescent staining of bovine (ovarian) granulosa cells (bGCs) with anti-UFL1 antibody. (**c**) Bovine ovaries were analyzed by immunohistochemistry.

**Figure 2 biomolecules-10-00260-f002:**
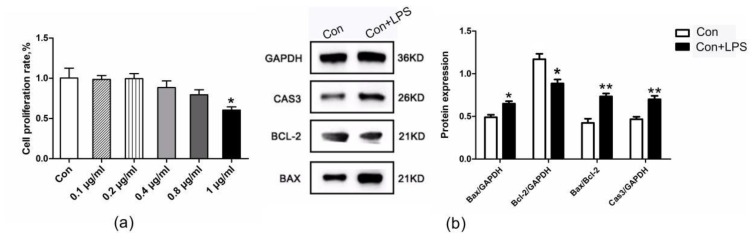
Lipopolysaccharide (LPS) treatment leads to increased apoptosis of ovarian granulosa cells. (**a**) Cell Counting Kit-8 (CCK-8) used to determine the effects of LPS on the cytotoxicity to GCs cells. (**b**) Effect of LPS (1 μg/mL) on the protein expression of BAX, BCL-2, and CAS3 in bGCs. Data are presented as the means ± the standard errors of the mean (SEM). ∗ *p* < 0.05; ∗∗ *p* < 0.01.

**Figure 3 biomolecules-10-00260-f003:**
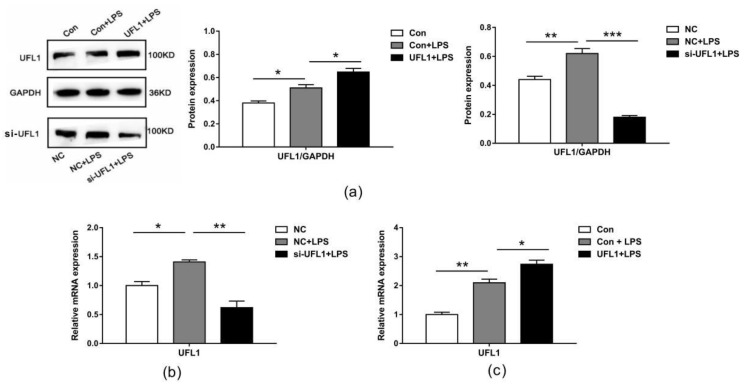
UFL1 is highly expressed in LPS-induced bGCs, and under LPS induction conditions, *Ufl1* small-interfering RNA (siRNA) significantly reduced UFL1 mRNA and protein expression; overexpression of UFL1 significantly increased UFL1 expression. (**a**) Effect of LPS and UFL1 siRNA/UFL1 overexpression plasmid on the protein expression of UFL1 in bGCs. (**b**) qPCR analysis of *Ufl1* in UFL1 siRNA-transfected bGCs stimulated with LPS. (**c**) qPCR analysis of *Ufl1* in bGCs transfected with *Ufl1* overexpression plasmid stimulated with LPS. Data are presented as the means ± the standard errors of the mean (SEM) of three independent experiments. ∗ *p* < 0.05; ∗∗ *p* < 0.01; ∗∗∗ *p* < 0.001.

**Figure 4 biomolecules-10-00260-f004:**
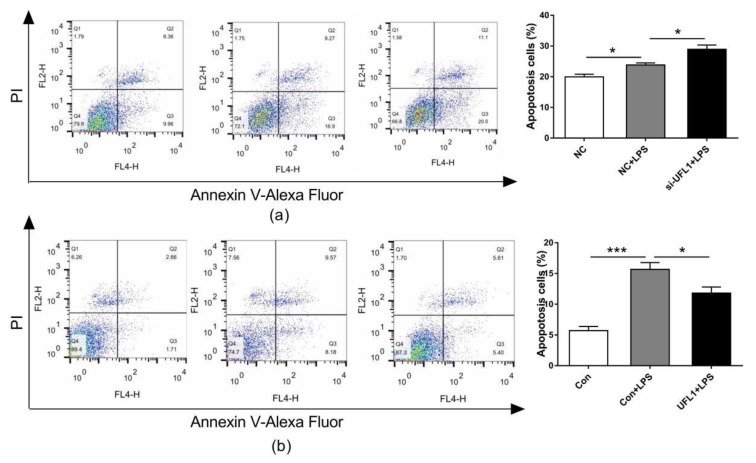
UFL1 protects the bGCs against LPS-induced apoptosis. (**a**) Effect of *Ufl1* siRNA on apoptosis in LPS-challenged bGCs. (**b**) Effect of *Ufl1* overexpression on apoptosis in LPS-challenged bGCs. Data are presented as the means ± the standard errors of the mean (SEM) of three independent experiments. ∗ *p* < 0.05; ∗∗ *p* < 0.01; ∗∗∗ *p* < 0.001.

**Figure 5 biomolecules-10-00260-f005:**
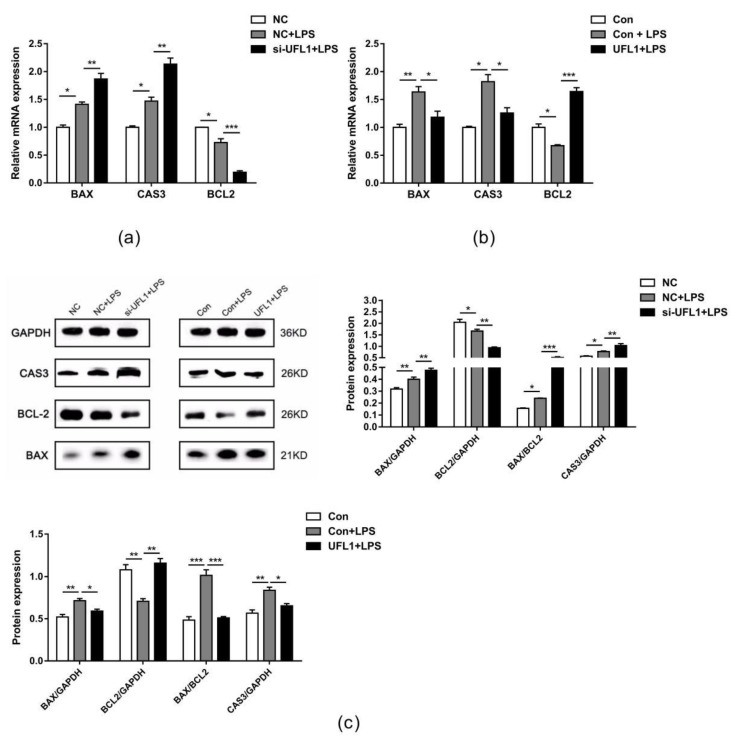
(**a**) Effect of *Ufl1* siRNA on activity of *Bax*, *Cas3*, and *Bcl-2* in LPS-challenged bGCs. (**b**) Effect of *Ufl1* overexpression on activity of *Bax, Cas3*, and *Bcl-2* in LPS-challenged bGCs. Data are presented as the means ± the standard errors of the mean (SEM) of three independent experiments. ∗ *p* < 0.05; ∗∗ *p* < 0.01; ∗∗∗ *p* < 0.001. (**c**) Representative Western blots and quantitative evaluation of BAX, CAS3, and BCL-2 in *Ufl1* siRNA-transfected bGCs and *Ufl1* overexpression plasmid stimulated with LPS.

**Figure 6 biomolecules-10-00260-f006:**
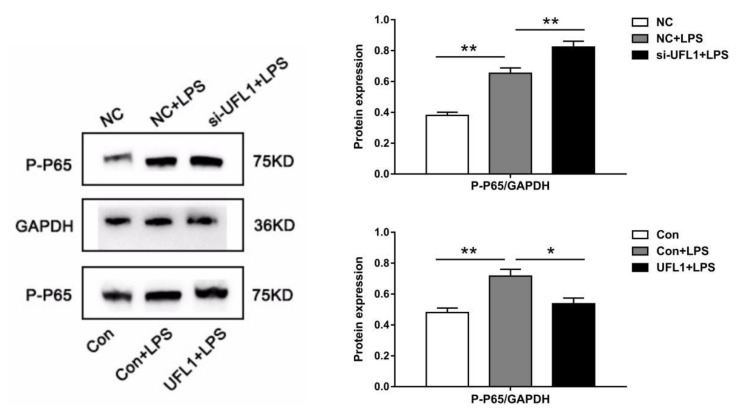
UFL1 inhibits phosphorylation of NF-κB p-p65. Effect of *Ufl1* siRNA and *Ufl1* overexpression plasmid on the protein expression of p-p65 in bGCs. Data are presented as the means ± the standard errors of the mean (SEM) of three independent experiments. ∗ *p* < 0.05; ∗∗ *p* < 0.01.

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
