# Peer review of "UFL1 Alleviates LPS-Induced Apoptosis by Regulating the NF-κB Signaling Pathway in Bovine Ovarian Granulosa Cells"

_biomolecules, 2020, doi:10.3390/biom10020260_

Round 1

Reviewer 1 Report

Present study shows that UFL1 is an regulator of LPS induced apoptosis in bovine granulosa cells and the effect is included in NFkB signaling pathway.

The paper is straightforward and easy understandable. My concern is about the lack of information and some in discussion as described below

L88 lps LPS

L101 what does mean “it was identified by Jinsirui ?

L101 what is the concentration of vector or siRNA for transfection

L101 what is the sequence of siRNA

L101 what is the construct of overexpression vector of UFL1

L108 were the results of RT-PCR normalized by that that of GAPDH?

L118 Granulosa cells depend on GAPDH. Do the LPS stimulation affect GAPDH expression levels in GCs?

L169 bCC bGC

L250 what is the in nature concentration of LPS in FF? Is 1 microg/ml higher or lower than that observed in FF?

L270-271 why can author conclude the effect of UFL1 observed in the present study is independent of its E3 ligase bactivity?

Figure 3 there was no explanation about 3-c

Figure 5 Please add the explanation of CAS3 (abbreviation) in figure legend

Reviewer 2 Report

The paper is interesting but I have many concerns :

It is not clear which ovarian follicles were punctioned ? from small, medium, large follicle. The granulosa cells from these different types of follicles have not the same ability to proliferate or to promote apoptosis The 1microg/ml dose of LPS appears very high. What is the physiological significativity ? The authors investigated the effects of UFL1 on bovine granulosa cells on cell viabilty and apoptosis but they should also determine the effect on steroidogenesis that is the main activity of these cells. The authors should analyse other signaling pathways that NF-kappaB

The english is poor, the paper needs an english editing

Author Response

Thank you for the reviewer’s valuable comments.

Round 2

Reviewer 2 Report

The authors answered to all the questions.